biomechanics

strength balance, hamstring strain injuries, isokinetic dynamometry, eccentric contraction

**Author for correspondence:**
Dimitrios Voukelatos
e-mail: D.Voukelatos@lboro.ac.uk

# The hamstrings to quadriceps functional ratio expressed over the full angle-angular velocity range using a limited number of data points

Dimitrios Voukelatos[1], Pavlos E. Evangelidis[2] and Matthew T. G. Pain[1]

[1]School of Sport, Exercise and Health Sciences, Loughborough University, Loughborough LE11 3TU, UK
[2]Faculty of Sport Sciences, Waseda University, Saitama 359-1192, Japan

 DV, 0000-0001-6589-3960; PEE, 0000-0002-2612-5173; MTGP, 0000-0002-0028-2365

The hamstring to quadriceps (H:Q) strength ratio is widely used to identify individuals at risk of sustaining hamstring strain injuries. However, its efficacy is not supported by the current evidence. Current methods for the calculation of the H:Q ratio provide only a one- or two-dimensional ratio, often ignoring fundamental muscle mechanical properties. Based on isokinetic torque measurements of the knee flexors and extensors ($0–400° \, s^{-1}$) in 25 young, physically active males, we derived a model equation that creates a three-dimensional H:Q functional ratio profile. The model robustness was tested against a different number of input torque data (8, 11, 14 and 17 pairs of points) and small perturbation of the knee joint angle data (5°). The model was consistent and behaved well under all conditions apart from the eight pairs of points ($R^2 = 0.84–0.96$; RMSE = 0.14–0.25; NRMSE = 0.12–0.27), and the H:Q functional ratio was successfully described even at angles and velocities that cannot be normally assessed with isokinetic dynamometry. Overall, our results suggest that the model can provide a fast and accurate three-dimensional description of the knee joint muscle strength balance using as few as 11 experimental data points and this could be an easy-to-employ screening tool.

## 1. Introduction

Hamstring strain injuries (HSI) are common in sprint-based sports, including different codes of football, track and field, field

hockey and baseball, and they are often sustained during acceleration and high-speed running leading to large losses in training and competition time [1–9]. Muscle strength imbalance around the knee joint has been long considered a risk factor for HSI [10] and is typically examined with the hamstring-to-quadriceps (H : Q) ratio [11]. The rationale is that knee joint control is determined primarily by the agonist–antagonist relationship between the hamstrings and quadriceps muscle groups. A lower H : Q ratio would suggest that hamstrings' strength capacity is not adequate to counter powerful quadriceps muscle actions and thus joint and/or muscle integrity may be compromised. Initially, the H : Q ratio was calculated by dividing the hamstrings peak concentric torque by the quadriceps peak concentric torque, known as the 'conventional' H : Q ratio, H : $Q_{con}$, [11]. However, this approach does not reflect the antagonist role of the hamstrings during knee extension which is to resist (i.e. eccentric action) the motions produced by the concentric contraction of the quadriceps. In the context of HSI during running, hamstrings decelerate the forward motion of the thigh and shank during the mid- and late-swing phase opposing the hip flexion and knee extension moment produced by the quadriceps [12–14]. Crucially, the late-swing phase is considered to be the most likely time of HSI [15]. To better represent the eccentric action of the hamstrings during knee extension the 'functional' H : Q ratio (H : $Q_{fun}$) was proposed, which is calculated by dividing the hamstrings eccentric peak torque with the quadriceps concentric peak torque at the same angular velocity of contraction [16–18]. Yet, any H : Q ratio based on peak torque fails to account for the changes in muscle force capacity due to changes in muscle length. To account for this, the angle-specific H : Q ratio has been introduced [18,19].

Despite numerous efforts to elucidate the association between knee joint strength balance and risk of HSI, current evidence remains inconclusive, with some studies supporting the use of the H : Q ratio to predict HSI [20–24] while others found no association [25–31]. A recent meta-analysis synthesized the evidence over a large variety of isokinetic strength measures—including H : $Q_{con}$ and H : $Q_{fun}$—and found that only slow velocity eccentric hamstring strength could have some limited potential to detect an athlete with increased risk of injury [32]. These contradictory results probably stem from the range of methods employed to assess strength imbalances at various combinations of velocities and modes of contraction as well as the varied correlation-based analyses used. Logistic regressions and odds ratios/risk ratios are most commonly implemented within the studies cited here, but $\chi^2$-tests, receiving operator curves and discriminant analysis have also been used. Different cut-off ratios have been found between studies using similar methods. Correlation techniques may be necessitated by the nature of studying hamstrings injury in humans but the lack of mechanistic research or causal experimental results is a limitation. In addition, the lack of a clearly defined normative cut-off H : Q value (i.e. the value below which the risk of hamstring injury increases significantly) further confounds whether knee joint strength imbalances predispose to HSI [31,33].

The lack of a universally accepted assessment protocol indicates that the choice of angular velocities at which the H : Q ratio is obtained is somewhat arbitrary. Also, as mentioned above, functional and conventional peak torque ratios are a function of angular velocity ($\omega$) only, whereas the effect of joint angle ($\theta$) is neglected. This oversight can greatly affect the measured H : Q ratio values as it rises significantly with increasing angle and angular velocity [19,34,35]. The general lack of angle-specific isokinetic torque measurements can be partly attributed to the difficulty in obtaining reliable torque data at obtuse angles, especially at high velocities [36]. Furthermore, any isokinetic measurement is limited to angular velocities well below those observed during high-speed running that can reach values over $1200° \, s^{-1}$ [37], reducing the ecological validity of these measurements.

Hiemstra *et al.* [34] created three-dimensional ($\omega$, $\theta$) dynamic control ratio maps of the H : $Q_{fun}$ ratio by dividing mean hamstrings' eccentric torque by the concentric quadriceps torque on a 'point by point basis'. This allowed for the effects of both $\omega$ and $\theta$ on the value of H : $Q_{fun}$ ratio to be simultaneously examined; however, maximum voluntary eccentric and concentric torque was recorded at isovelocities that did not exceed $250° \, s^{-1}$, whereas the scalar output of the maps precluded the extrapolation to angular velocities and angles beyond the measured range. In order for the hamstrings–quadriceps interaction during knee extension to be fully described, the H : $Q_{fun}$ ratio needs to be expressed as a function of both $\omega$ and $\theta$. Ideally, this function should be capable of calculating the H : $Q_{fun}$ at high extension angles and angular velocities. Since such measurements are not feasible with the isokinetic dynamometers, a model equation with the above characteristics would be a very useful tool in the study of the H : $Q_{fun}$ ratio and its association to hamstrings' pathology.

The aim of this work was to derive an equation that will describe the functional H : Q torque ratio as a function of two variables, namely angular velocity $\omega$ and angle $\theta$. Ideally, this function (henceforth termed $R_E(\omega, \theta)$) should have a small enough number of parameters to be determined quickly and efficiently thus requiring few ($\omega, \theta$) points. At the same time, it should be accurate enough to provide a sufficient qualitative and quantitative description of the functional H : Q ratio at knee joint angles that cannot, normally, be attained during isokinetic contractions of the hamstrings and quadriceps muscles especially at angular velocities of over $300° \, s^{-1}$.

# 2. Method

## 2.1. Experimental data

Raw torque-angular velocity-angle ($T$-$\omega$-$\theta$) datasets were used from two previously published studies by the authors [35,38]. The first dataset [38] included measurements of knee extensor and flexor muscles' isometric and isokinetic torque at 10 isovelocities. This dataset was used to determine a suitable function to represent the H : Q ratio (2.1) (detailed below). The second dataset [35] was obtained from a different cohort and it was used to test the goodness of fit of the newly developed function and compare it with the respective values from the first dataset. The experimental procedures for these data collections have been detailed elsewhere [35,38]. Briefly, for the first dataset, 11 healthy male subjects completed maximum voluntary contractions (MVC) for isometric, concentric and eccentric knee extensions and flexions on an isokinetic dynamometer [38]. Isometric measurements were obtained at five angles of joint flexion that spanned the subject's joint range of motion (ROM). Isokinetic maximum voluntary eccentric-concentric cycle contractions were performed at ten different angular velocities, $\pm 50$, 100, 200, 300, $400° \, s^{-1}$. For the second dataset, a different cohort of 14 healthy male subjects performed a very similar protocol but with isokinetic contractions obtained at only $\pm 60$, 240, $400° \, s^{-1}$ [35]. In both studies, to account for human and machine compliance, crank angles and crank angular velocities were converted to joint angles and joint angular velocities using a linear regression equation derived from digitized joint and crank angle data collected during a subset of trials of each subject [35,38]. An overview of the methods used in this paper is presented in figure 1.

## 2.2. Derivation of $R_T(\omega, \theta)$

The first step in the derivation of $R_E(\omega, \theta)$ was to obtain a description of the behaviour of the H : Q$_{fun}$ ratio with concurrently varying $\theta$ and $\omega$ by means of a theoretical three-dimensional H : Q ratio function termed $R_T(\omega, \theta)$. The purpose of $R_T(\omega, \theta)$ was to function as a benchmark for $R_E(\omega, \theta)$, providing information on its mathematical properties and behaviour. For clarity, the variable coefficients of $R_T(\omega, \theta)$ and $R_E(\omega, \theta)$ will, henceforth, be called *parameters* and will be determined by fitting the function to a set of the *dependent variables* $\omega$ and $\theta$.

The theoretical H : Q$_{fun}$ ratio function, $R_T(\omega, \theta)$, was based on the following piecewise function as previously described in Forrester *et al.* [39]

$$T^{MVC}(\omega, \theta) = \begin{cases} T^{tet}_{conc}(\omega)\alpha(\omega)T(\theta), & \omega \geq 0 \\ T^{tet}_{ecc}(\omega)\alpha(\omega)T(\theta), & \omega < 0 \end{cases} \tag{2.1}$$

$T^{MVC}$ is a mathematical function that expresses the maximum voluntary torque output of a muscle as a function of the joint angular velocity of contraction $\omega$ and the joint angle $\theta$ for both the concentric and eccentric phases of contraction. Equation (2.1) is a nine-parameter function based on underlying muscle physiological properties that provides a three-dimensional description of the subject's specific theoretical torque profile (figure 2a,b). This function has been used multiple times to represent the variation in maximal voluntary joint torque as a function of angular velocity and angle with good results [39–42]. $T^{tet}_{conc}(\omega)$ and $T^{tet}_{ecc}(\omega)$ are two rectangular hyperbolas representing tetanic force output as a function of contraction velocity in concentric ($\omega \geq 0$) and eccentric ($\omega < 0$) contractions, respectively [40]. $\alpha(\omega)$ and $T(\theta)$ represent the level of muscle activation and joint angle, respectively. Expanding into the concentric and eccentric phases gives

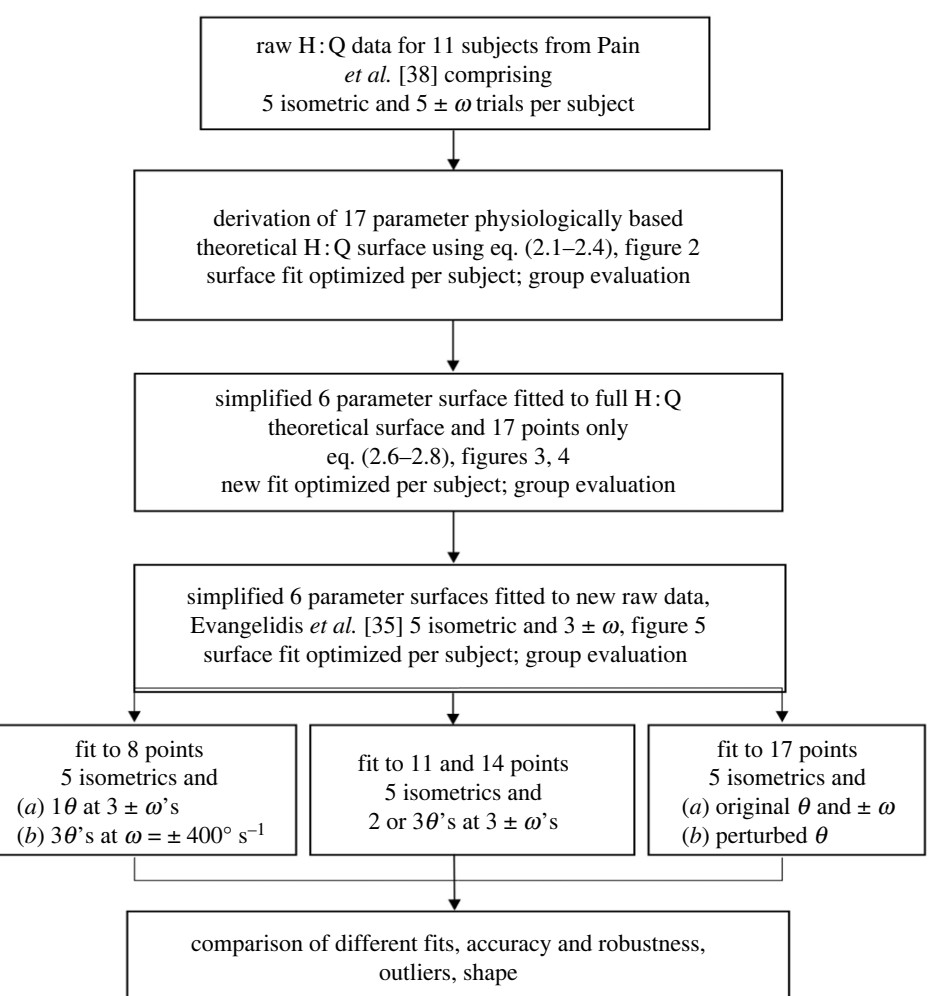

**Figure 1.** Flow chart of the key steps for the derivation of $R_E(\omega, \theta)$; $\omega$ is angular velocity and $\theta$ is knee joint angle. In each of the fits five of the points were taken from isometric contractions while the remaining points were taken from different combinations of angles and angular velocities (no more than three angular velocities were used).

— $\omega \geq 0$

$$T_{conc}^{tet}(\omega, \theta) = \left[\left(\frac{T_0 w_c (w_{max} + w_c)}{w_{max}(w_c + w)} - \frac{T_0 w_c}{w_{max}}\right)\right]$$
$$\times \left(\alpha_{min} + \frac{\alpha_{max} - \alpha_{min}}{1 + e^{(-((\omega - \omega_1)/\omega_r))}}\right) e^{(-(1/2))\left[(\theta - \theta_{opt})^2/W^2\right]} \tag{2.2}$$

— $\omega < 0$

$$T_{ecc}^{tet}(\omega, \theta) = -\left[\frac{(T_{max} - T_0)^2 \omega_{max}\omega_c}{\kappa T_0(\omega_{max} + \omega_c)[(((T_{max} - T_0)\omega_{max} + \omega_c)/\kappa T_0(\omega_{max} + \omega_c)) - \omega]} + T_{max}\right]$$
$$\times \left(\alpha_{min} + \frac{\alpha_{max} - \alpha_{min}}{1 + e^{(-((\omega - \omega_1)/\omega_r))}}\right) e^{(-(1/2))\left[(\theta - \theta_{opt})^2/W^2\right]}, \tag{2.3}$$

where $T_{max}$ is the maximum eccentric torque, $T_0$ is the maximum isometric torque, $\omega_{max}$ is the maximum angular velocity, $\omega_c$ is the vertical asymptote of the concentric hyperbola (equation 2.2), $\omega_r$ is the angular velocity range over which the ramp-up in differential activation takes place, $\alpha_{min}$ is the low plateau activation level and $\omega_1$ is the angular velocity at the midpoint of the $\alpha(\omega)$ versus $\omega$ ramp, $\theta_{opt}$ is the optimal angle for torque production and $W$ is the width (standard deviation) of the curve.

For the derivation of the theoretical H : $Q_{fun}$ ratio function, $R_T(\omega, \theta)$, the dataset of Pain *et al.* [38] was used as it provides the value of knee joint torque at 10 different isovelocities, which allows a more

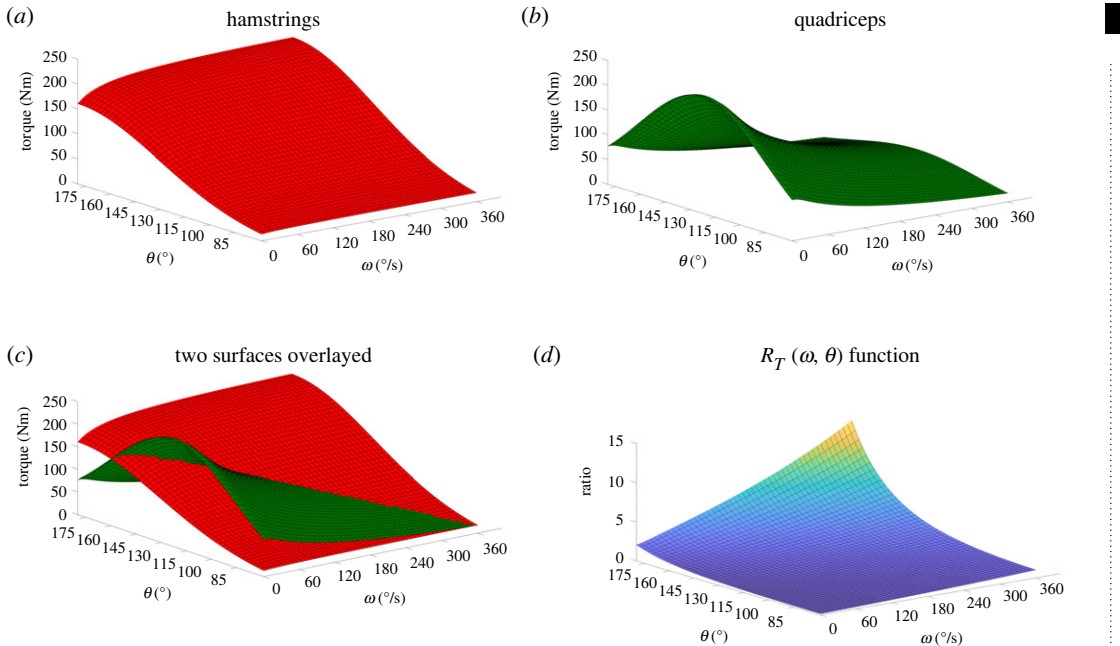

**Figure 2.** $T_{\text{ecc}_{\text{H}}}^{\text{MVC}}(\omega, \theta)$ for hamstrings and $T_{\text{conc}_{\text{Q}}}^{\text{MVC}}(\omega, \theta)$ for quadriceps shown separately (a,b) and overlayed (c). Their quotient, the $R_T(\omega, \theta)$ function, is demonstrated in (d).

accurate description of the $T$-$\omega$ relationship. The nine parameters described above were optimized individually for each one of the 11 $T$-$\omega$-$\theta$ datasets using the simulated annealing algorithm of Corana *et al.* [43] where the parameter values are varied within bounds in order to minimize the root mean square difference between $T^{\text{MVC}}(\omega, \theta)$ and experimental torques [39]. Surfaces were optimized on a per subject basis rather than on pooled group data, as it has previously been shown that subject-specific torque parameters are needed, not group averages, to represent performances based on torque output [44]. Group data were pooled for output goodness of fit scores for statistical analysis.

Following the determination of a $T^{\text{MVC}}(\omega, \theta)$ function from the hamstrings and quadriceps $T$-$\omega$-$\theta$ datasets of each participant the theoretical H : $Q_{\text{fun}}$ ratio function, $R_T(\omega, \theta)$ was obtained using the equation

$$R_T(\omega, \theta) = \frac{T_{\text{ecc}_{\text{H}}}^{\text{MVC}}(\omega, \theta)}{T_{\text{conc}_{\text{Q}}}^{\text{MVC}}(\omega, \theta)}, \tag{2.4}$$

where $T_{\text{ecc}_{\text{H}}}^{\text{MVC}}(\omega, \theta)$ was obtained from hamstrings eccentric, and $T_{\text{conc}_{\text{Q}}}^{\text{MVC}}(\omega, \theta)$ was obtained from quadriceps concentric contraction, respectively (figure 2a–c). This results in a 17-parameter function as there are nine parameters for the eccentric mode of contraction and eight for the concentric mode (figure 2d).

## 2.3. Derivation of the experimental $R_E(\omega, \theta)$ function

Once the theoretical ratio surfaces given by $R_T(\omega, \theta)$ had been obtained, the descriptive $R_E(\omega, \theta)$ ratio function that would have the ability to accurately reproduce the physiologically derived $R_T(\omega, \theta)$ function using fewer parameters was determined. This involved creating plane curves of $R_T(\omega, \theta)$ by setting first $\omega$ and then $\theta$ equal to a constant value $c$ from the ratio surface of a single subject.

$$\left.\begin{aligned} R_T^c(\theta) &= R_T(c, \theta) \\ R_T^c(\omega) &= R_T(\omega, c), \end{aligned}\right\} \tag{2.5}$$

and

where $c = 0, 60, 180, 400°\,\text{s}^{-1}$ for $R_T(c, \theta)$ and $c = 0, 30, 60, 75°$ for $R_T(\omega, c)$. Subsequently different functions, and linear combinations of them, were fitted to the plane curves using Matlab (The MathWorks Inc., Natick, MA) for different values of $c$ to determine the one that gave the best fit. Goodness of fit was measured via the coefficient of determination, $R^2$, and the root mean square error (RMSE). It was found that $R_T^c(\theta)$ and $R_T^c(\omega)$ were best described by the following three- and four-parameter functions, $R_E(\theta)$ and $R_E(\omega)$, respectively (figure 3).

$$R_E^c(\theta) = a_1\,\mathrm{e}^{a_2\theta} + a_3 \tag{2.6}$$

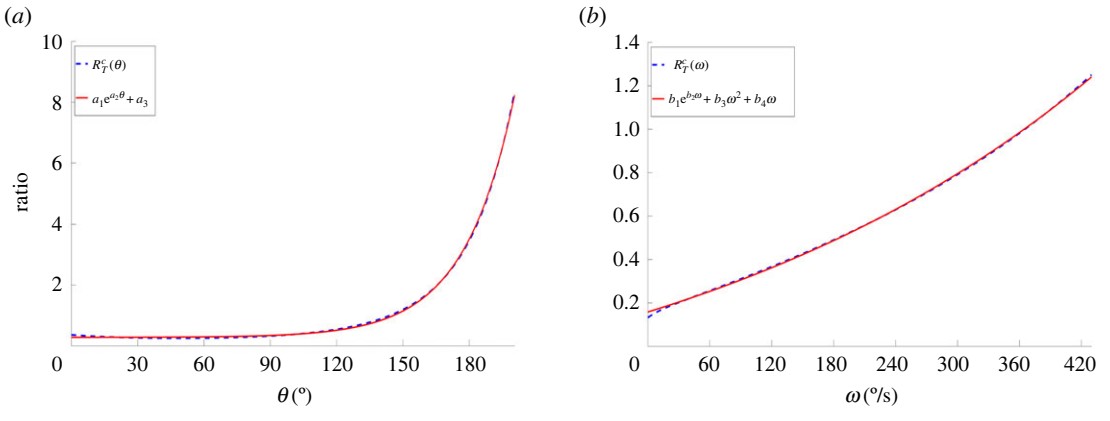

**Figure 3.** Derivation of $R_E(\theta)$ and $R_E(\omega)$ by fitting three- and four-parameter functions on the plane curves $R_T(c, \theta)$ and $R_T(\omega, c)$ of the $R_T(\omega, \theta)$ function. (a) $R_T^c(\theta)$ and (b) $R_T^c(\omega)$.

and

$$R_E^c(\omega) = b_1\, \mathrm{e}^{b_2\omega} + b_3\omega^2 + b_4\omega \tag{2.7}$$

as these produced the best fits for all values of $c$ (table 2).

Equations (2.6) and (2.7) indicated that the two-variable $R_E(\omega, \theta)$ function would need to contain a combination of exponential and polynomial terms with respect to both $\omega$ and $\theta$. Consequently, various linear, e.g. $R_E^c(\theta) + R_E^c(\omega)$, and nonlinear, e.g. $\omega \cdot R_E^c(\theta) + \theta \cdot R_E^c(\omega)$ or $R_E^c(\theta) \cdot R_E^c(\omega)$, combinations of $R_E^c(\theta)$ and $R_E^c(\omega)$ were fitted on the $R_T(\omega, \theta)$ surface in order to determine $R_E(\omega, \theta)$. The best fit was obtained by the six-parameter exponential function

$$R_E(\omega, \theta) = a\exp\left(b\omega^n + c\theta^m\right) - \mathrm{d}\omega^{1/2}\theta^2 \tag{2.8}$$

To test the ability of the $R_E(\omega, \theta)$ function to accurately reproduce the $R_T(\omega, \theta)$, the function was first fitted individually to each of the 11 $R_T(\omega, \theta)$ surfaces using the whole surface for the fit. Subsequently, to establish if it could accurately do so using a limited number of data points only 17 $(\omega, \theta, R_T(\omega, \theta))$ points from each of the 11 theoretical $R_T(\omega, \theta)$ ratio surfaces were chosen and $R_E(\omega, \theta)$ was fitted per subject to those points and compared with the whole surface fits. Seventeen $(\omega, \theta, R_T(\omega, \theta))$ points were used for the fit as the minimum number of points required to define each of $R_T(\omega, \theta)$ surfaces. Goodness of fit was assessed via the $R^2$ and RMSE values. All surface fits were performed in Matlab using least squares. All coefficients were given a lower bound of zero when $R_E(\omega, \theta)$ was fitted to the $R_T(\omega, \theta)$ surface; however, coefficients $b$ and $c$ were allowed to obtain negative values when fitted to the raw data points.

## 2.4. Testing the $R_E(\omega, \theta)$ function on raw H : Q$_{\mathrm{fun}}$ ratio values

Fitting equation (2.8) on $R_T(\omega, \theta)$ is not a conclusive enough test as it is an analytically defined function and therefore not susceptible to the experimental errors that would be encountered in an actual testing environment. For this reason, we tested the goodness of fit of $R_E(\omega, \theta)$ when fitted on the second raw experimental dataset of Evangelidis *et al.* [35]. This dataset provided fewer $T(\omega)$ points than the first dataset (6 v. 10), and therefore a less detailed description of the $T$-$\omega$ relationship; however, its data collection protocol is simpler and requires less time to perform, therefore making it more likely to be used during testing.

The first step was to ascertain that the goodness of the fit on the theoretical surfaces was not dataset-dependent. To establish this, the same testing process applied on the first dataset was used. Specifically, $R_T(\omega, \theta)$ surfaces were calculated for each one of the 14 subjects of the new dataset and the $R_E(\omega, \theta)$ function was fitted first to the whole surface and then to 17 $(\omega, \theta, R_T(\omega, \theta))$ points from each surface. Group $R^2$ and RMSE values were obtained and compared with the respective values from the first dataset.

Next, we sought to determine how well the $R_E(\omega, \theta)$ function would be able to reproduce the experimental H : Q$_{\mathrm{fun}}$ values and to assess its sensitivity to the number of points used for the fit. This was done in two stages. During the first stage the experimental, raw, H : Q functional ratios, $R_{\mathrm{exp}}$, at 11, 14 and 17 $(\omega, \theta)$ points were calculated individually for every one of the 14 subjects and $R_E(\omega, \theta)$ was fitted on every $(\omega, \theta, R_{\mathrm{exp}})$ set of points, (figure 5). Each one of the 11-, 14- and 17-point sets consisted of five $R_{\mathrm{exp}}$ points calculated during isometric contractions $(\omega = 0, \theta, R_{\mathrm{exp}})$. The 11-point

**Table 1.** Synopsis of the different $R_{exp}$ ratio points used in the six fits. Number of $R_{exp}$ points calculated during isometric ($\omega = 0$) and isovelocity trials ($\omega \neq 0$), number of different isovelocities employed, $m$ and number of different joint angles per isovelocity, $n$.

| no. points fitted | no. points per isovelocity $\omega$ | | | | value of $m$ | value of $n$ |
|---|---|---|---|---|---|---|
| | $0°\ s^{-1}$ | $60°\ s^{-1}$ | $240°\ s^{-1}$ | $400°\ s^{-1}$ | | |
| 8 ($\omega_{max}$, $\theta$, $R_{exp}$) | 5 | 0 | 0 | 1 | 1 | 3 |
| 8 ($\omega$, $\theta_{max}$, $R_{exp}$) | 5 | 1 | 1 | 1 | 3 | 1 |
| 11 ($\omega$, $\theta$, $R_{exp}$) | 5 | 2 | 2 | 2 | 3 | 2 |
| 14 ($\omega$, $\theta$, $R_{exp}$) | 5 | 3 | 3 | 3 | 3 | 3 |
| 17 ($\omega$, $\theta$, $R_{exp}$) | 5 | 4 | 4 | 4 | 3 | 4 |
| 17 ($\omega$, $\theta_{varied}$, $R_{exp}$) | 5 | 4 | 4 | 4 | 3 | 4 |

version had an additional six points made up of two points at each of the three isovelocities, measured at the maximum and minimum knee joint angles. In addition the 14-point set included one extra ($\omega$, $\theta$, $R_{exp}$) point per isovelocity calculated at 50% of joint ROM while the 17-point set included two extra ($\omega$, $\theta$, $R_{exp}$) points per isovelocity compared to the 11-points, calculated at 33% and 66% of knee joint ROM (table 1).

Having established the goodness of fit values of the $R_E(\omega, \theta)$ function on both theoretical and raw $R_{exp}$ points, further tests were performed to determine whether it would be possible (i) to further reduce the number of raw ratio points used in the fit and (ii) to determine how sensitive $R_E(\omega, \theta)$ would be to discrepancies in the values of the knee joint angle $\theta$ where torque is measured during hamstrings and quadriceps contractions. This was done to establish the behaviour of the $R_E(\omega, \theta)$ function when isovelocity data is provided only at a single joint angle or single isovelocity. These two options represent two different experimental testing modes that could be considered as some of the quickest and easiest to perform while getting isovelocity data. A single isovelocity is easy to collect experimentally but then some data processing for the three angles is needed. Three isovelocities, but only at maximum angle, is more time-consuming experimentally but this could probably be read from on screen information during collection. To achieve this, $R_E(\omega, \theta)$ was fitted on the following sets of raw, $R_{exp}$ points.

— An eight-point raw ratio set ($\omega$, $\theta$, $R_{exp}$) consisting of five ratio points calculated during isometric contraction ($\omega = 0$, $\theta$, $R_{exp}$) and three ($\omega$, $\theta$, $R_{exp}$) points calculated at the maximum value of knee joint angle ($\theta_{max}$) for each of the isovelocities ($\omega$, $\theta_{max}$, $R_{exp}$).
— An eight-point raw ratio set ($\omega_{max}$, $\theta$, $R_{exp}$) consisting of five ($\omega = 0$, $\theta$, $R_{exp}$) ratio points and three ($\omega$, $\theta$, $R_{exp}$) points calculated at the maximum isovelocity $400°\ s^{-1}$.
— A 'varied' 17-point raw ratio set ($\omega$, $\theta_{varied}$, $R_{exp}$) consisting of five ($\omega = 0$, $\theta$, $R_{exp}$) ratio points and 12 ($\omega$, $\theta$, $R_{exp}$) points, four per isovelocity, where the hamstring $\theta$ values were offset to be 5° lower than the respective values for quadriceps.

As before, the accuracy of the fit was assessed using the $R^2$ and RMSE values. Furthermore, to assess the error between different fits the normalized RMSE scores (NRMSE) were calculated by dividing the respective RMSE scores by the range values. All algebraic calculations were performed using MAPLE 16 (Maplesoft Inc., Waterloo, Ontario, Canada).

# 3. Results

## 3.1. Derivation of the experimental $R_E(\omega, \theta)$ function

Fitting equations (2.6) and (2.7) on the plane curves $R_E(\theta)$ and $R_E(\omega)$, produced mean $R^2$ values of 0.99 for both fits and mean RMSE values of $0.08 \pm 0.06$ and $0.005 \pm 0.001$ (mean ± s.d.) for the $R_E(\theta)$ and $R_E(\omega)$ fits, respectively (table 2).

## 3.2. Fit of $R_E(\omega, \theta)$ function on $R_T(\omega, \theta)$

The six-parameter $R_E(\omega, \theta)$ function was fitted to 17 ($\omega$, $\theta$, $R_T(\omega, \theta)$) ratio points and the whole $R_T(\omega, \theta)$ surface, and the goodness of the fits assessed using the $R^2$ and RMSE values are summarized in table 3.

**Table 2.** $R^2$ and RMSE values for the fit of equations (2.6) and (2.7) on $R_T^c(\theta)$ and $R_T^c(\omega)$, respectively.

| | $R_T^c(\theta)$ | | | $R_T^c(\omega)$ | |
| $c$ (° s$^{-1}$) | $R^2$ | RMSE | $c$ (°) | $R^2$ | RMSE |
| --- | --- | --- | --- | --- | --- |
| 0 | 0.99 | 0.02 | 0 | 0.99 | 0.006 |
| 60 | 0.99 | 0.04 | 30 | 0.99 | 0.005 |
| 180 | 0.99 | 0.07 | 60 | 0.99 | 0.005 |
| 400 | 0.99 | 0.18 | 75 | 0.99 | 0.004 |

**Table 3.** Mean and standard deviation (s.d.), $R^2$ and RMSE values for the fit of the $R_E(\omega, \theta)$ function on the theoretical $R_T(\omega, \theta)$ ratio surface, and on 17 points of that surface for both datasets.

| mean ± s.d. | | |
| | $R^2$ | RMSE |
| --- | --- | --- |
| [a]17-point fit | 0.99 ± 0.001 | 0.05 ± 0.05 |
| [a]Whole surface fit | 0.99 ± 0.003 | 0.06 ± 0.08 |
| [b]17-point fit | 0.99 ± 0.02 | 0.04 ± 0.05 |
| [b]Whole surface fit | 0.98 ± 0.03 | 0.04 ± 0.05 |

[a]Datasets from Pain *et al.* [38].
[b]Datasets from Evangelidis *et al.* [35].

The 17-point fit produced mean $R^2$ and RMSE values that were very similar to the $R^2$ and RMSE values exhibited by the whole surface fit of the $R_E(\omega, \theta)$ function on the theoretical $R_T(\omega, \theta)$, for both datasets (table 3).

## 3.3. Fit of $R_E(\omega, \theta)$ function on raw H : Q$_{fun}$ points

### 3.3.1. 8-, 11-, 14- and 17-point fits

The fit of the $R_E(\omega, \theta)$ function on 8, 11, 14 and 17 ($\omega$, $\theta$, $R_{exp}$) points of the second $T$-$\omega$-$\theta$ dataset produced mean $R^2$ values that although high appeared to decline as the number of points used in the fit increased, ranging between 0.84 and 0.96. On the contrary, the respective mean RMSE and NRMSE values did not exhibit the same trend with values ranging between 0.14–0.25 and 0.12–0.27, respectively (table 4).

Across the 14 subjects, there was a spread of $R^2$, RMSE and NRMSE scores and it was noted that in some cases the shape of the curve could substantially deviate from the theoretical shape. To check for outliers a 14-choose-10 combinations was run on the RMSE values separately for 11-, 14- and 17-point fits. The combinations with the lowest RMSE score always had four of the same five subjects' data missing. These combinations were 2.79 to 3.35 s.d. from the mean of all other combinations. The worst four had means across the 11-, 14- and 17-point conditions that were between 3.69 and 6.41 s.d. from the group mean and were removed as outliers, and the group data was recalculated (table 4).

## 4. Discussion

This study aimed to derive a model equation for the three-dimensional H : Q$_{fun}$ ratio profile in terms of $\omega$ and $\theta$, and this was successfully accomplished with the six-parameter function $R_E(\omega, \theta)$, which used as few as 11 experimental data points, obtainable with only five isometric and three isovelocity trials on an isokinetic dynamometer. Fits using 11, 14 or 17 experimental data points all produced high $R^2$ and low RMSE values, indicating that $R_E(\omega, \theta)$ behaved consistently in producing a description of the H : Q$_{fun}$ ratio. Fits to only eight points had much higher NRMSE values and some of the surface shapes were very different from the theoretical profile. However, tests with the perturbed 17 points with $\theta$ offset

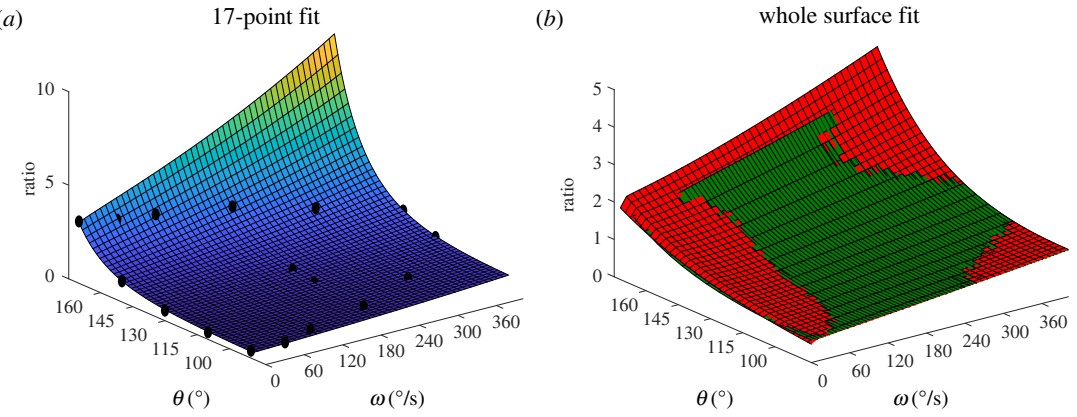

**Figure 4.** (a) Example of fitting $R_E(\omega, \theta)$ on 17 $(\omega, \theta, R_T(\omega, \theta))$ points. (b) Example of fitting $R_E(\omega, \theta)$ (in red) on the whole $R_T(\omega, \theta)$ surface (in green).

**Table 4.** Mean $R^2$, RMSE and NRMSE values for all six fits and subjects and for the 11-, 14- and 17-point fits with subjects 2, 8, 9, 10 excluded.

| | mean ± s.d. | | |
| --- | --- | --- | --- |
| no. points fitted | $R^2$ | RMSE | NRMSE |
| 8 ($\omega_{max}$, $\theta$, $R_{exp}$) | 0.96 ± 0.04 | 0.21 ± 0.14 | 0.27 ± 0.15 |
| 8 ($\omega$, $\theta_{max}$, $R_{exp}$) | 0.96 ± 0.06 | 0.15 ± 0.19 | 0.24 ± 0.21 |
| 11 ($\omega$, $\theta$, $R_{exp}$) | 0.91 ± 0.10 | 0.23 ± 0.16 | 0.12 ± 0.06 |
| 14 ($\omega$, $\theta$, $R_{exp}$) | 0.87 ± 0.07 | 0.25 ± 0.11 | 0.13 ± 0.03 |
| 17 ($\omega$, $\theta$, $R_{exp}$) | 0.84 ± 0.11 | 0.25 ± 0.13 | 0.13 ± 0.03 |
| 17 ($\omega$, $\theta_{varied}$, $R_{exp}$) | 0.89 ± 0.04 | 0.14 ± 0.03 | 0.15 ± 0.03 |
| [c]11 ($\omega$, $\theta$, $R_{exp}$) | 0.96 ± 0.02 | 0.14 ± 0.05 | 0.09 ± 0.03 |
| [c]14 ($\omega$, $\theta$, $R_{exp}$) | 0.89 ± 0.05 | 0.20 ± 0.08 | 0.13 ± 0.03 |
| [c]17 ($\omega$, $\theta$, $R_{exp}$) | 0.89 ± 0.04 | 0.19 ± 0.06 | 0.12 ± 0.02 |

[c]Mean $R^2$, RMSE and NRMSE values for the 11-, 14- and 17-point fits with subjects 2, 8, 9, 10 excluded.

by 5° between flexors and extensors, demonstrated that the $R_E(\omega, \theta)$ $R^2$ values were not affected by small perturbations in the values of $\theta$.

The benchmark 17-parameter $R_T(\omega, \theta)$ function represented the H : $Q_{fun}$ increasing with angle and angular velocity as would be expected from both theory and experimental results. This function was based on a nine-parameter torque function that has been used multiple times to represent maximal voluntary joint torque as a function of angle and angular velocity with good results [39–42] and therefore it should be suitable to act (i) as a starting point for the derivation of a simpler ratio function, $R_E$ and (ii) as a benchmark against which the new function is tested.

When the six-parameter exponential-based function $R_T(\omega, \theta)$ was fitted first on the whole $R_T(\omega, \theta)$ surfaces and subsequently on 17 points of the $R_E(\omega, \theta)$ surface it was very successful in reproducing the original surfaces as shown by the high correlation (table 3 and figure 4). When $R_E(\omega, \theta)$ was fitted on the second dataset its quantitative accuracy for the 11-, 14- and 17-point fits was also good and there was very little difference between the results of the 11 ($\omega$, $\theta$, $R_{exp}$) points fit and those of the 14 and 17 ($\omega$, $\theta$, $R_{exp}$) points fits (table 4 and figure 5b). It should be noted that, with a single exception, all three fits were qualitatively consistent i.e. if the 11-point fit predicted an increase in H : $Q_{fun}$ values with increasing angular velocity then that trend was repeated in the other two fits (figure 5b). The only exceptions were the 11- and 14-point fits for Subject 2 that extrapolated to a low value of H : $Q_{fun}$ contrary to the 17-point fit that predicted a high H : $Q_{fun}$ value for large $\omega$ and $\theta$ (figure 5a). When the number of points was reduced to eight, for both versions the initial $R^2$ and RMSE values were

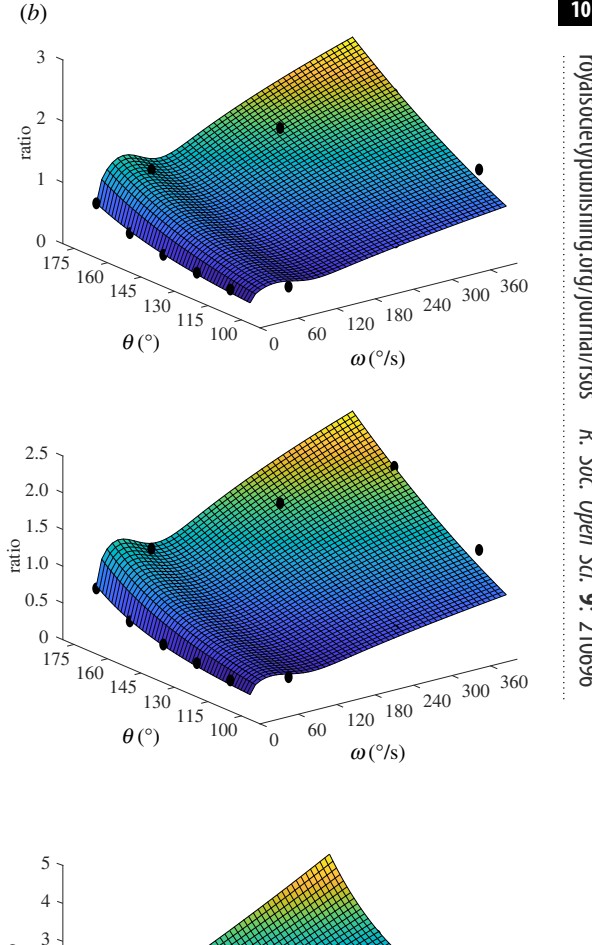

(*a*)

(*b*)

**Figure 5.** Fit of the $R_E(\omega, \theta)$ function on 11 (top row), 14 (middle row) and 17 (bottom row) $(\omega, \theta, R_{exp})$ points of the raw H : Q surface for Subjects 2 and 4. The ratio values of the 11- and 14-point fits for Subject 2 fail to increase with increasing $\omega$ and $\theta$ values, due to an abnormally high ratio value at $(400°\ s^{-1}, 107°)$. (*a*) Subject 2 and (*b*) Subject 4.

comparable to the 11-, 14- and 17-point fits, but the NRMSE values were almost twice as large. There was a much greater range in surface shapes, with more deviating from the theoretical shape and some predicting negative ratio values, whereas all measured values are positive. Paradoxically the $R^2$ values for these were good, as even when the surface was unrealistic, the $(\omega, \theta, R_{exp})$ points still fell on it as they were calculated near the extremes of $\omega$ and $\theta$ ranges with no mid-range points to drag the middle of the surface above zero.

Closer inspection of the second dataset revealed that four subjects were outliers, which resulted from their surfaces deviating from the theoretical shape, table 4. These four subjects attained their maximum ratio at either low angular velocities or at low angles of extension, $(\omega, \theta)$ values of $(60°\ s^{-1}, 159°)$, $(400°\ s^{-1}, 113°)$, $(400°\ s^{-1}, 107°)$ and $(400°\ s^{-1}, 108°)$, whereas both previous experimental studies [17–19,45–48] and the theoretical ratio function $R_T(\omega, \theta)$ show that the ratio value should increase with increasing $\omega$ and $\theta$ as a consequence of the T-$\omega$ and T-$\theta$ relationships (figure 4). A poor $R^2$ score may be indicative of an individual's inability to produce maximal torque during hamstrings eccentric contraction or of sub-maximal effort on the part of the subject during quadriceps concentric contraction which may lead to erroneous results (figure 5*a*). Voluntary contractions, especially eccentric and slow concentric muscle actions, may be affected by reduced neural drive despite maximal effort from the participants [49]. While this would affect the fit of the T-$\omega$-$\theta$ surface, it may be less evident in standard peak torque

ratios. Nevertheless, both causes (i.e. inability to produce maximal torque and sub-maximal efforts) can be important in the case of a prospective study where the $H : Q_{fun}$ ratio is used as a predictor for possible future hamstrings injuries. The former because it would allow the identification of injury-susceptible athletes and the latter because sub-maximal quadriceps contractions at an angle where they are expected to be stronger than the hamstrings might artificially increase the $H : Q_{fun}$ ratio above the 0.8–1.05 cut-off range often used [18,22,23,31] confounding the ratio's potential (if any) to predict HSI. Fitting the $R_E(\omega, \theta)$ function on the isovelocity data can provide an early indication of any such issue (figure 5). Where the $R^2$ score of the fit is low or the fit deviates from the theoretical shape, it is likely that those discrepancies were caused either by weak hamstrings muscles or from potential abnormalities in the measurements.

There is also further evidence to suggest that the $R_E(\omega, \theta)$ function could be used as a screening and rehabilitation tool. Hiemstra *et al.* [34], used ratio maps consisting of 2500 ($\omega, \theta, R_{exp}$) points to compare subjects with anterior cruciate ligament (ACL) repairs with healthy controls and found significant differences in the $H : Q_{fun}$ ratio values of patients that had undergone ACL reconstruction compared to healthy subjects (1.35 versus 1.57, respectively) at relatively high angular velocities and extension angles. They, too, hypothesized that this type of 'regional strength changes' cannot be picked up by the $H : Q_{fun}$ for the same reasons discussed here.

The $R_E(\omega, \theta)$ function provides a more complete description of the $H : Q_{fun}$ ratio and may facilitate its use as a screening tool to predict at-risk athletes. While the $H : Q_{fun}$ ratio was proposed, and has been used repeatedly, as a better descriptor of the knee joint strength balance than the $H : Q_{con}$ ratio, its use for injury prediction is not supported by current evidence [32,50]. As can be seen in the $R_E(\omega, \theta)$ function (figures 4 and 5), there is no single ratio value but a well-defined surface of values, thus it is less surprising that previous studies using single values and specific cut-offs, have not had comparable and consistent outcomes. It may well prove informative to examine previous $H : Q_{fun}$ ratio studies and establish whether the $R_E(\omega, \theta)$ function could be used to normalize between them in some way.

Further considerations for the use of the $H : Q_{fun}$ ratio in general can also be informed from having ratio values from the $R_E(\omega, \theta)$ function at obtuse angles of extension, in joint positions where the hamstrings are under high strain and therefore susceptible to injury [51–53]. For example, the mean $H : Q_{fun}$ ratio evaluated on the 14 subjects at a higher velocity at near full extension ($400° \, s^{-1}$, $172°$), gave a mean $R_E$ value of 2.73, which is notably higher than any of the cut-off values that have been reported in the literature as possible injury predictors. While this higher value is to be expected given the nature of the $H : Q_{fun}$ ratio surface, it highlights the testing-specific nature of a single $H : Q_{fun}$ ratio value, and how it necessitates correlating it with injuries within a cohort to obtain a useful value that may well be very study-specific. It also demonstrates that with the joint angle more extended and velocities high, in what would be considered conditions where the muscle would be more susceptible to injury, the hamstrings are comfortably stronger than the quadriceps. Near full extension the quadriceps have all but lost their ability to contract concentrically and it is not solely their action as agonists that the hamstrings are contracting eccentrically to counteract. Therefore, the $H : Q_{fun}$ ratio, of any type, should not be viewed as a mechanical threshold between the torque that can be generated by the hamstrings and quadriceps under some single angle or velocity condition, which is then correlated with an injury risk. It should be considered as the hamstrings' ability to develop sufficient torque to safely counteract the extensor angular momentum vector acting about the knee when nearing full extension, and how this ability correlates with an injury risk. As torque causes change in angular momentum, the higher the capacity for quadriceps torque development, throughout its range, the greater the angular momentum that could be developed as near full extension is reached, even though the instantaneous torque generating capacity of the quadriceps may now be low. However, it should also be noted that other factors, besides the often dominant concentric action of the quadriceps, contribute to the gain in angular momentum about the knee, such as the hip extension motion and passive transfers of power between the torso and the thigh, as well as the mass of the shank + foot segment [13,54]. This may explain the relative success of the $H : Q_{con}$ ratio as a hamstring injury predictor, as it gives an indirect estimate of the absolute quadriceps' strength. Indeed, a low conventional ratio may not necessarily mean weak hamstrings, but strong quadriceps [21,50,55]. The simple peak torque of the quadriceps and the hamstrings at their own optimal angle, probably gives a good indication of the overall capacity of each muscle group to do work, which is needed to control the angular accelerations and decelerations about the knee joint. Nevertheless, the $H : Q_{con}$ ratio cannot account for any neural effects, such as the aforementioned reduction in neural drive during fast eccentric

muscle contractions, that could affect the ability of the hamstrings to successfully decelerate the lower leg.

A benefit of using this surface method includes not relying on only one or two effectively instantaneous points per muscle group to get a ratio which, irrespective of contraction conditions, only gives a limited view of the muscle's work/strength capacity over its functional range. If neural effects in the eccentric phase are contributing to a relative hamstring weakness this will not be evident with H : $Q_{con}$, as there is no eccentric hamstrings measure, and may easily be missed in H : $Q_{fun}$ if the eccentric hamstrings value(s) did not get measured during a period of inhibition, or reduced activation for any reason. Even if the (single-point) H : Q ratio is examined at selected angles where injuries are expected to occur, alongside high dynamometer velocities, these angles may not cover those where the reduced neural contributions may be having a deleterious effect. In addition, a H : $Q_{fun}$ ratio measured at near full extension will be inherently high, as was shown earlier, and even with reduced activation could be high and surpass the currently applied cut-off values. Another possibility is that neural activation is high at optimal muscle lengths for both the quadriceps and the hamstrings, as this is within the best operating lengths of the muscles, and so the typical H : $Q_{fun}$ ratio will not include neural limitations. However, if activation was reduced at some intermediate angle as the hamstrings are entering their period of deceleration, they would be unable to do enough work to safely decelerate the shank in the remaining time/distance, and they would be exposed to increased risk of injury even with now high levels of activation and a high H : $Q_{fun}$ ratio.

A major limitation with isokinetic dynamometry is the restricted angular velocity and ROM that can be examined, especially compared with the ranges seen in running where hamstrings injuries occur. However, the $R_E(\omega, \theta)$ function provides angle-specific estimates of the H : $Q_{fun}$ ratio at more obtuse angles of extension and at higher velocities than can be measured directly. The simplified form of the $R_E(\omega, \theta)$ compared to the full $R_T(\omega, \theta)$ method means that obtaining the $R_{exp}$ ratio values is a relatively simple process that can be readily performed after data collection. Thus, $R_E(\omega, \theta)$ values can be calculated in a single testing session even by a non-expert. Considering that the alternative would be to follow the multi-step process described in equations (2.1)–(2.3), employing the $R_E(\omega, \theta)$ function not only significantly reduces the need for extensive isokinetic testing protocols but also the time needed for data processing. It is worth noting that five of the total points used in the fit correspond to isometric measurements ($\omega = 0$). These points are the easiest to obtain experimentally and offer the added benefit of a high test-retest reliability [56]. Six further ratio points were used (in the case of the 11-point fit) to obtain the $R_E(\omega, \theta)$ values, two each from isokinetic measurements at 60, 240 and 400° s$^{-1}$. Using five isometric points and two points at equally spaced angles from each of the three different isovelocities is recommended to give the best compromise between goodness of fit and minimal data points. This allows the smoothly curving theoretical shape to be produced when points across the measurements are well-behaved yet still allows for a divergence from this with an additional peak or trough or a negative surface gradient, for increasing $\omega$ and $\theta$, when some points are erroneous. The 14- and 17-point fits just used more points from the three velocities, which increases processing time slightly but not data collection time. The evaluation of the function requires a single Matlab or Excel script, which allows its use by non-specialist staff during the testing session or even when only manual recording and calculation of H : Q values is feasible.

To conclude, in this study, the functional H : $Q_{fun}$ ratio was described as a function of $\omega$ and $\theta$, $R_E(\omega, \theta)$. The function was fitted to a variety of experimentally obtained ratio points and its accuracy was assessed by means of $R^2$, RMSE and NRMSE values. The function was robust to changes in the number of points used in the fit and exhibited consistent results across all fits. Although not an original aim of this study it was found that obtaining a shape for the whole surface that adheres to the fundamental muscle mechanical properties, rather than a value at any single point on the surface, seems to provide more insight into potential oddities with the subject's relative hamstrings and quadriceps performance. This limits the dependence on any one set of numbers in order to determine the hamstrings to quadriceps function. Those results showed that the $R_E(\omega, \theta)$ function can provide a fast and accurate description of the three-dimensional H : $Q_{fun}$ ratio profile of a test subject from 11 data points without them having to undergo extensive and time-consuming isovelocity tests.

Ethics. All participants completed physical activity and health screen questionnaires before providing written informed consent for their participation. Both study protocols were approved by the Loughborough University Ethical Approvals (Human Participants) Sub-Committee (under permit nos. R11-P102 and R12-P119). Informed consent was obtained from all participants.

Data accessibility. The code that fits the ratio function to the raw torque dataset has been uploaded as part of the electronic supplementary material along with a proxy dataset [57]. The ethical approval awarded by the author's institution for

this study permits data to be publicly accessible only if this is explicitly declared on the subject information sheet and consent form. As this was not the case in this study, making the data publicly accessible would not conform to the licence the authors have been granted to use this data. With the agreement of the journal's Editorial Office, the authors will not be able to make the dataset available on this occasion, but encourage readers, referees and editors to contact the Ethics Approvals Sub-Committee at Loughborough University for data access requests at ssehs-res-ent@lboro.ac.uk.

Authors' contributions. D.V.: conceptualization, data curation, formal analysis, methodology, project administration, software, validation, visualization, writing—original draft, writing—review and editing. P.E.E.: conceptualization, data curation, investigation, methodology, resources, validation, visualization, writing—original draft, writing—review and editing; M.T.G.P.: conceptualization, investigation, methodology, resources, supervision, validation, visualization, writing—original draft, writing—review and editing.

All authors gave final approval for publication and agreed to be held accountable for the work performed therein.
Conflict of interest declaration. We declare we have no competing interests.
Funding. No funding has been received for this article.

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
