## [Peer Review File · Royal Society Open Science]

Review History

RSOS-210696.R0 (Original submission)

Review form: Reviewer 1

Is the manuscript scientifically sound in its present form?

No

Are the interpretations and conclusions justified by the results?

Yes

Is the language acceptable?

Yes

Do you have any ethical concerns with this paper?

No

Have you any concerns about statistical analyses in this paper?

Yes

Recommendation?

Major revision is needed (please make suggestions in comments)

Comments to the Author(s)

This is an interesting paper that aims find a H:Q ratio equation that could be used to profile athlete's torque generation capacity and potentially predict hamstring injuries.

Although the work done can be seen as slightly incremental, the authors have tried to improve the current body of literature by using a combination of datasets and newly collected data to validate their computational work.

As better explained here below, there are some major issues related to current version of the paper:

1. The methods sections should be restructured and written more plainly as at the moment it is very difficult to follow. For example the steps used to calculate the Re and Rt functions (and the final identification for the best parameters of the plane curves) could be summarised and better explained using a figure.
2. The rationale for using a specific dataset for model implementation and then model validation should be discussed and clarified. The author should consider to pool all the data and randomly identify the training and validation datasets or even perform a cross-validation with all data. This is key to provide the best fit possible and better evaluate the model robustness.
3. The minimum number of points need to fit the curves should be identified using an optimisation.
4. The fit evaluation using different number of points generates rather different curves with predicted values for high angles and velocities that can vary dramatically. This can generate much bigger errors than the RMSE reported and also provide wrong prediction for hamstring injuries during high velocity tasks. This point should be addressed.

Line 46-57. Although this is a well written paragraph more info on the type of analysis used to predict hamstring injury using such metrics should be clarified. If those studies only featured correlation approaches, also the methods should be listed as a potential limitation.

METHODS

Personally, I would include the experimental methods first and use subheadings to split them from the theoretical ones.

Line 91-122. This is a rather difficult paragraph to follow without reading any previous work, and the use of Re and Rt is a bit confusing. Perhaps a figure here to show the different steps and meaning of parameters would help.

The parameter optimised should be more clearly highlighted to avoid confusion with the ones inputted from experimental measurements.

Line 136 - Clarify how angular velocities were converted. At the moment the wording makes it unclear.

Line 205. I believe that a cross-validation approach should be embrace to evaluate the group-dependency. It is unclear if this was done only once.

Line 230-241. This paragraph is very difficult to follow and quite cryptic. Table 1 surely helps. Also why did not the authors also run on optimisation to find the set of points needed for the best fit?

Line 273-278. I do not believe this is a rigorous way to carry out a cross-validation.

Figure 4. A figure including all data points and the curves should be included. It is very confusing to see only a single subject data point.

Table 4. Why is a ~ 0.20 acceptable? What happens at higher speeds and velocities?

Decision letter (RSOS-210696.R0)

Dear Dr Voukelatos

The Editors assigned to your paper RSOS-210696 "A three-dimensional description of the Hamstrings:Quadriceps functional ratio" have now received comments from reviewers and would like you to revise the paper in accordance with the reviewer comments and any comments from the Editors. Please note this decision does not guarantee eventual acceptance.

Please submit your revised manuscript and required files (see below) no later than 21 days from today's (ie 22-Oct-2021) date. Note: the ScholarOne system will 'lock' if submission of the revision is attempted 21 or more days after the deadline. If you do not think you will be able to meet this deadline please contact the editorial office immediately.

Kind regards,
Royal Society Open Science Editorial Office

on behalf of Dr Jonas Rubenson (Associate Editor) and Pietro Cicuta (Subject Editor)
openscience@royalsociety.org

Associate Editor Comments to Author (Dr Jonas Rubenson):

Comments to the Author:

Dear Dr. Voukelatos and co-authors,

As you will see, the review is overall positive. Nevertheless, it raises some concerns that you will need to address before this work can be considered for publication. Specifically, Reviewer 1 raises important concerns regarding the description of the methodology that needs to be addressed.

I share similar concerns regarding the representation of the study approach and methodology. In particular, it was not clear from the outset that the study involves both a meta-analysis of previous data collected and published by the authors, as well as a new experimental data set.

This can be clarified. Also, there is not very much information on the new experimental methodology. In fact, as far as I understood, there is more detailed information provided for the previously published experiment. For example, there is no information on the subject characteristics or the specific experimental procedures that were performed in the new experiments. Also, if studies were performed on human subjects, information on human ethics approval should be documented.

Minor points:

Lines 69-81. From this paragraph, it is not immediately clear how the study of Hiemstra et al that uses omega and theta is a 2D analysis but your study also using omega and theta is considered a 3D analysis.

Lines 131-156. There is a lot of detailed information provided on the methodology that has been previously published. This could be simplified, and more detail could instead be provided for the new experiments performed.

Line 438. "accurate enough" is subjective. Consider rewording. If the conclusion is that it is an accurate approach, it could simply be stated as such.

Besides the methodological concerns, I also feel that the study can provide a valuable new tool to help practitioners assess the H:Q ratio. As such, I hope that these concerns can be addressed in a revision.

All the best,
Jonas Rubenson

Reviewer comments to Author:

Reviewer: 1

Comments to the Author(s)

This is an interesting paper that aims find a H:Q ratio equation that could be used to profile athlete's torque generation capacity and potentially predict hamstring injuries.

Although the work done can be seen as slightly incremental, the authors have tried to improve the current body of literature by using a combination of datasets and newly collected data to validate their computational work.

As better explained here below, there are some major issues related to current version of the paper:

1. The methods sections should be restructured and written more plainly as at the moment it is very difficult to follow. For example the steps used to calculate the R_e and R_t functions (and the final identification for the best parameters of the plane curves) could be summarised and better explained using a figure.
2. The rationale for using a specific dataset for model implementation and then model validation should be discussed and clarified. The author should consider to pool all the data and randomly identify the training and validation datasets or even perform a cross-validation with all data. This is key to provide the best fit possible and better evaluate the model robustness.
3. The minimum number of points need to fit the curves should be identified using an optimisation.
4. The fit evaluation using different number of points generates rather different curves with predicted values for high angles and velocities that can vary dramatically. This can generate much bigger errors than the RMSE reported and also provide wrong prediction for hamstring injuries during high velocity tasks. This point should be addressed.

Line 46-57. Although this is a well written paragraph more info on the type of analysis used to predict hamstring injury using such metrics should be clarified. If those studies only featured correlation approaches, also the methods should be listed as a potential limitation.

METHODS

Personally, I would include the experimental methods first and use subheadings to split them from the theoretical ones.

Line 91-122. This is a rather difficult paragraph to follow without reading any previous work, and the use of R_e and R_t is a bit confusing. Perhaps a figure here to show the different steps and meaning of parameters would help.

The parameter optimised should be more clearly highlighted to avoid confusion with the ones inputted from experimental measurements.

Line 136 - Clarify how angular velocities were converted. At the moment the wording makes it unclear.

Line 205. I believe that a cross-validation approach should be embraced to evaluate the group-dependency. It is unclear if this was done only once.

Line 230-241. This paragraph is very difficult to follow and quite cryptic. Table 1 surely helps. Also why did not the authors also run on optimisation to find the set of points needed for the best fit?

Line 273-278. I do not believe this is a rigorous way to carry out a cross-validation.

Figure 4. A figure including all data points and the curves should be included. It is very confusing to see only a single subject data point.

Table 4. Why is a ~ 0.20 acceptable? What happens at higher speeds and velocities?

===PREPARING YOUR MANUSCRIPT===

===PREPARING YOUR REVISION IN SCHOLARONE===

-- If you have uploaded ESM files, please ensure you follow the guidance at <https://royalsociety.org/journals/authors/author-guidelines/#supplementary-material> to include a suitable title and informative caption. An example of appropriate titling and captioning may be found at https://figshare.com/articles/Table_S2_from_Is_there_a_trade-off_between_peak_performance_and_performance_breadth_across_temperatures_for_aerobic_sc_ope_in_teleost_fishes_/3843624.

Author's Response to Decision Letter for (RSOS-210696.R0)

See Appendix A.

RSOS-210696.R1 (Revision)

Review form: Reviewer 1

Is the manuscript scientifically sound in its present form?

No

Are the interpretations and conclusions justified by the results?

Yes

Is the language acceptable?

Yes

Do you have any ethical concerns with this paper?

Yes

Have you any concerns about statistical analyses in this paper?

No

Recommendation?

Accept with minor revision (please list in comments)

Comments to the Author(s)

The methods of the paper have been more clearly described. However the authors could put a bit more effort in making figure 1 more palatable (i.e. by adding respective graphs) and easier to understand.

The conclusion should clearly provide guidelines on the number of points (or experiments) that should be done to use the equation correctly. This does not come up clearly from the conclusion and should be included in the abstract too.

Also I feel that the title could be improve. The key message is to be able to use few experimental points as inputs to a novel equation to estimate H:Q ratio. The '3-D dimensional' description is not very helpful to me.

SPECIFIC COMMENTS

Line 98 - Please avoid the use of acronyms in this first sentence as it hinders the readability.

Line 102-103. What do the authors mean with 'test the robustness'? Is this a validation? If so what's the acceptable error and why?

Figure 1 - I am afraid but I do not find Figure 1 very explanatory. Also there is no mention of 'training dataset' and 'validation/testing dataset'. Perhaps the inclusion on some graphs taken for the other paper figures might help.

Line 194-195. How did the authors vary the combinations? What was the rationale for moving from linear to non-linear? I think this needs more explanation.

Line 199 Perhaps I am missing something but how can the authors test the best fit using the same 11 subject? Should not this be tested on the 14 subject database? Perhaps the authors are referring to testing the robustness of the 17 points?

Line 198-201. I am sorry but this paragraph is still too cryptic.

Line 207 - I think should be equation 8.

Line 462. How can the reader know what's the correct shape? I guess the reader would know what are the inputs (i.e. number of points) needed to use the proposed formula?

Decision letter (RSOS-210696.R1)

Dear Dr Voukelatos

On behalf of the Editors, we are pleased to inform you that your Manuscript RSOS-210696.R1 "A three-dimensional description of the Hamstrings:Quadriceps functional ratio" has been accepted for publication in Royal Society Open Science subject to minor revision in accordance with the referees' reports. Please find the referees' comments along with any feedback from the Editors below my signature.

Please submit your revised manuscript and required files (see below) no later than 7 days from today's (ie 09-Mar-2022) date. Note: the ScholarOne system will 'lock' if submission of the revision is attempted 7 or more days after the deadline. If you do not think you will be able to meet this deadline please contact the editorial office immediately.

on behalf of Dr Jonas Rubenson (Associate Editor) and Pietro Cicuta (Subject Editor)
openscience@royalsociety.org

Associate Editor Comments to Author (Dr Jonas Rubenson):

Comments to the Author:

Dear Dr. Voukelatos and co-authors,

As you will see, the review is generally positive. However, R1 raises some concerns that should be addressed. In particular, the reviewer's suggestions for Figure 1 and their comments about the concluding remarks can help improve your manuscript. I hope that you find the reviewer comments helpful I look forward to seeing a revised manuscript.

Best Regards,
Jonas Rubenson

Reviewer comments to Author:

Reviewer: 1

Comments to the Author(s)

The methods of the paper have been more clearly described. However the authors could put a bit more effort in making figure 1 more palatable (i.e. by adding respective graphs) and easier to understand.

The conclusion should clearly provide guidelines on the number of points (or experiments) that should be done to use the equation correctly. This does not come up clearly from the conclusion and should be included in the abstract too.

Also I feel that the title could be improve. The key message is to be able to use few experimental points as inputs to a novel equation to estimate H:Q ratio. The '3-D dimensional' description is not very helpful to me.

SPECIFIC COMMENTS

Line 98 - Please avoid the use of acronyms in this first sentence as it hinders the readability.

Line 102-103. What do the authors mean with 'test the robustness'? Is this a validation? If so what's the acceptable error and why?

Figure 1 - I am afraid but I do not find Figure 1 very explanatory. Also there is no mention of 'training dataset' and 'validation/ testing dataset'. Perhaps the inclusion on some graphs taken for the other paper figures might help.

Line 194-195. How did the authors vary the combinations? What was the rationale for moving from linear to non-linear? I think this needs more explanation.

Line 199 Perhaps I am missing something but how can the authors test the best fit using the same 11 subject? Should not this be tested on the 14 subject database? Perhaps the authors are referring to testing the robustness of the 17 points?

Line 198-201. I am sorry but this paragraph is still too cryptic.

Line 207 - I think should be equation 8.

Line 462. How can the reader know what's the correct shape? I guess the reader would know what are the inputs (i.e. number of points) needed to use the proposed formula?

===PREPARING YOUR MANUSCRIPT===

one version should clearly identify all the changes that have been made (for instance, in coloured highlight, in bold text, or tracked changes);

===PREPARING YOUR REVISION IN SCHOLARONE===

- If you are providing image files for potential cover images, please upload these at this step, and inform the editorial office you have done so. You must hold the copyright to any image provided.
- A copy of your point-by-point response to referees and Editors. This will expedite the preparation of your proof.

- Ensure that your data access statement meets the requirements at <https://royalsociety.org/journals/authors/author-guidelines/#data>. You should ensure that you cite the dataset in your reference list. If you have deposited data etc in the Dryad repository, please only include the 'For publication' link at this stage. You should remove the 'For review' link.
- If you are requesting an article processing charge waiver, you must select the relevant waiver option (if requesting a discretionary waiver, the form should have been uploaded, see 'File upload' above).
- If you have uploaded any electronic supplementary (ESM) files, please ensure you follow the guidance at <https://royalsociety.org/journals/authors/author-guidelines/#supplementary-material> to include a suitable title and informative caption. An example of appropriate titling and captioning may be found at https://figshare.com/articles/Table_S2_from_Is_there_a_trade-off_between_peak_performance_and_performance_breadth_across_temperatures_for_aerobic_scope_in_teleost_fishes_/3843624.

Author's Response to Decision Letter for (RSOS-210696.R1)

See Appendix B.

Decision letter (RSOS-210696.R2)

Dear Dr Voukelatos,

It is a pleasure to accept your manuscript entitled "The Hamstrings:Quadriceps functional ratio expressed over the full angle-angular velocity range using a limited number of data points" in its current form for publication in Royal Society Open Science.

on behalf of Dr Jonas Rubenson (Associate Editor) and Pietro Cicuta (Subject Editor)
openscience@royalsociety.org

Appendix A

Associate Editor Comments to Author (Dr Jonas Rubenson):

Comments to the Author:

Dear Dr. Voukelatos and co-authors,

As you will see, the review is overall positive. Nevertheless, it raises some concerns that you will need to address before this work can be considered for publication. Specifically, Reviewer 1 raises important concerns regarding the description of the methodology that needs to be addressed.

I share similar concerns regarding the representation of the study approach and methodology. In particular, it was not clear from the outset that the study involves both a meta-analysis of previous data collected and published by the authors, as well as a new experimental data set. This can be clarified. Also, there is not very much information on the new experimental methodology. In fact, as far as I understood, there is more detailed information provided for the previously published experiment. For example, there is no information on the subject characteristics or the specific experimental procedures that were performed in the new experiments. Also, if studies were performed on human subjects, information on human ethics approval should be documented.

Authors' response: *Thank you for your constructive comments. We have now amended the manuscript accordingly to clarify our methodology.*

Minor points:

Lines 69-81. From this paragraph, it is not immediately clear how the study of Hiemstra et al that uses omega and theta is a 2D analysis but your study also using omega and theta is considered a 3D analysis.

Authors' response: *We agree that Hiemstra et al. used a 3D analysis. It used a 2D figure and in cutting down the text we have made this error of referring to the analysis as 2D. We have amended the text as follows (Line 75):*

“Hiemstra et al. [34] created 3-dimensional (3D) (ω, θ) dynamic control ratio maps...”

Lines 131-156. There is a lot of detailed information provided on the methodology that has been previously published. This could be simplified, and more detail could instead be provided for the new experiments performed.

Authors' response: *We have now restructured the Methods section considerably, providing only a brief overview of the raw data acquisition and referring the interested reader to our relevant published studies for a detailed description.*

Line 438. “accurate enough” is subjective. Consider rewording. If the conclusion is that it is an accurate approach, it could simply be stated as such.

Authors' response: *We agree with this comment. We have now removed the word “enough” and the text now reads (Line 466):*

“Those results showed that the $R_E(\omega, \theta)$ function can provide a fast and accurate description of the 3-dimensional $H:Q_{fun} \dots$ ”

Reviewer comments to Author:

Reviewer: 1

Comments to the Author(s)

This is an interesting paper that aims find a H:Q ratio equation that could be used to profile athlete's torque generation capacity and potentially predict hamstring injuries.

Although the work done can be seen as slightly incremental, the authors have tried to improve the current body of literature by using a combination of datasets and newly collected data to validate their computational work.

Authors' response:

Thank you for your useful comments and constructive critique of our work. We would like to clarify that this modelling work uses two experimental - raw - datasets previously collected by two of the authors of this manuscript. The first dataset (Pain et al.), which covers a wider range of angular velocities, was used to optimise the parameters of equation (1), while the second dataset (Evangelidis et al.) with fewer angular velocities but with more subjects was used to examine the robustness of equation (8). This has now been clarified in the revised Methods section (Lines 98-118).

As better explained here below, there are some major issues related to current version of the paper:

1. The methods sections should be restructured and written more plainly as at the moment it is very difficult to follow. For example, the steps used to calculate the R_e and R_t functions (and the final identification fo the best parameters of the plane curves) could be summarised and better explained using a figure.

Authors' response:

We have now restructured the Methods section considerably to make it clearer and easier to follow. We have also added a flow chart of the main steps and added some additional information on the surface optimisation being per subject in a few key places.

2. The rationale for using a specific dataset for model implementation and then model validation should be discussed and clarified. The author should consider to pool all the data and randomly identify the training and validation datasets or even perform a cross-validation with all data. This is key to provide the best fit possible and better evaluate the model robustness.

Authors' response:

We have now expanded more on the rationale behind the choice of datasets for each stage of the model derivation and implementation. Specifically for the choice of the first dataset (lines: 154-164)

“For the derivation of the theoretical H:Q_{fun} ratio function, $R_T(\omega, \theta)$, the dataset of Pain et al. [38] was used as it provides the value of knee joint torque at ten different isovelocities which allows a more accurate description of the T– ω relationship. The

9 parameters described above were optimised individually for each one of the 11 T - ω - θ datasets using the Simulated Annealing algorithm of Corana et al. [43] where the parameter values are varied within bounds in order to minimise the root mean square difference between $T^{MVC}(\omega, \theta)$ and experimental torques [39]. Surfaces were optimized on a per subject basis rather than on pooled group data as it has previously been shown that subject specific torque parameters are needed, not group averages, to represent performances based on torque output [44]. Group data were pooled for output goodness of fit scores for statistical analysis.”

And on the choice of the second dataset (lines: 209-214)

“...For this reason, we tested the goodness of fit and robustness of $R_E(\omega, \theta)$ when fitted on the second raw experimental dataset of Evangelidis et al. [35]. This dataset provided fewer $T(\omega)$ points than the first dataset (6 vs 10) but on and therefore a less detailed description of the T - ω relationship. However, it contained data on more subjects, and its data collection protocol is simpler and requires less time to perform, therefore making it more likely to be used during testing”.

3. The minimum number of points need to fit the curves should be identified using an optimisation.

Authors' response:

It was not our intention to identify the absolute minimum number of points needed for the fit of the function but rather to test the function on a number of points that could be achieved in a standard testing protocol by non-research staff. However, we do recognise that we may have worded some parts as such as we did so implicitly with the practicality aspect in mind. We have removed reference to minimal/minimum number of points and we have added some short bits of text in places to mention the relationship with the experimental protocols and practicality.

In terms of actually looking for the true minimal number of points the minimum number of points would be 6 (two per isovelocity, at the endpoints of the joint ROM). If this minimum number of points then produced poor fits it would then become a case of looking at extra points that reduce the error due to noise in the data to some acceptable level. As it was 8 points were too few to provide a reliable fit, at least with the spread of the 8 points chosen. If we had just looked for the true minimum with any spread, such as could be taken from the theoretical surface it would likely have wanted good uniform spread of points and this would then entail specific isovelocities and angles in testing and we think this is a slightly different problem to solve.

4. The fit evaluation using different number of points generates rather different curves with predicted values for high angles and velocities that can vary dramatically. This can generate much bigger errors than the RMSE reported and also provide wrong prediction for hamstring injuries during high velocity tasks. This point should be addressed.

Authors' response:

We have added more discussion on the shape of the surface fit being an important consideration not just any point on it or even just the overall goodness of fit values.

Line 328-335

“When the number of points was reduced to 8, for both versions the initial R^2 and RMSE values were comparable to the 11-, 14-, and 17-point fits, but the NRMSE values were almost twice as large. There was a much greater range in surface shapes with more deviating from the theoretical shape and some predicting negative ratio values whereas all measured values are positive. Paradoxically the R^2 values for these were good as even when the surface was unrealistic the (ω , θ , R_{exp}) points still fell on it as they were calculated near the extremes of ω and θ ranges with no midrange points to drag the middle of the surface above zero.”

Line 461-465

“Although not an original aim of this study it was found that obtaining the correct shape for the surface rather than the value at any point on the surface seems to provide more insight into potential oddities with the subjects relative hamstrings and quadriceps performance. This limits the dependence on any one set of numbers in order to determine hamstrings to quadriceps function.”

Line 46-57. Although this is a well written paragraph more info on the type of analysis used to predict hamstring injury using such metrics should be clarified. If those studies only featured correlation approaches, also the methods should be listed as a potential limitation.”

Authors' response:

The cited papers did use various correlation approaches of varying complexity. We have added a section on the typical types of methods used and the limitations of this (lines 52-60).

“These contradictory results likely stem from the range of methods employed to assess strength imbalances at various combinations of velocities and modes of contraction as well as the varied correlation-based analyses used. Logistic regressions and odds ratios/risk ratios are most commonly implemented within the studies cited here, but chi-squared tests, receiving operator curves and discriminant analysis have also been used. Different cut off ratios have been found between studies using similar methods. Correlation techniques may be necessitated by the nature of studying hamstrings injury in humans but the lack of mechanistic research or causal experimental results is a limitation”

METHODS

Personally, I would include the experimental methods first and use subheadings to split them from the theoretical ones.

Authors' response: *Thank you for your suggestion. We have now included the experimental methods first as suggested (Lines 97-116).*

Line 91-122. This is a rather difficult paragraph to follow without reading any previous work, and the use of R_e and R_t is a bit confusing. Perhaps a figure here to show the different steps and meaning of parameters would help.

The parameter optimised should be more clearly highlighted to avoid confusion with the ones inputted from experimental measurements.

Authors' response: *This paragraph has been re-written to make more self-contained and highlight the variables optimised through simulated annealing (lines 119-139 and 154-164, we haven't included the relevant sections for brevity).*

Line 136 - Clarify how angular velocities were converted. At the moment the wording makes it unclear.

Authors' response: *This was amended and now reads (lines 112-116)*

“In both studies to account for human and machine compliance crank angles and crank angular velocities were converted to joint angles and joint angular velocities using a linear regression equation derived from digitised joint and crank angle data collected during a subset of trials of each subject [35, 38].”

Line 205. I believe that a cross-validation approach should be embraced to evaluate the group-dependency. It is unclear if this was done only once.

Authors' response: *We have performed the test of R_T for all the 14 subjects of the 2nd dataset (instead of selecting 3). Both 17 point and full surface fits have been performed and results are reported in Table 3 (lines 215-221).*

“The first step was to ascertain that the goodness of the fit on the theoretical surfaces was not dataset-dependent. To establish this the same testing process applied on the first dataset was used. Specifically, $R_T(\omega, \theta)$ surfaces were calculated for each one of the 14 subjects of the new dataset and the $R_E(\omega, \theta)$ function was fitted first to the whole surface and then to 17 ($\omega, \theta, R_T(\omega, \theta)$) points from each surface. Group R^2 and RMSE values were obtained and compared to the respective values from the first dataset.”

Line 230-241. This paragraph is very difficult to follow and quite cryptic. Table 1 surely helps. Also why did not the authors also run on optimisation to find the set of points needed for the best fit?

Authors' response: *This paragraph has now been re-written*

(Lines 227-233): “Each one of the 11-, 14- and 17-point sets consisted of five Rexp points calculated during isometric contractions ($\omega = 0, \theta, R_{exp}$). The 11-point version

had an additional six points made up of two points at each of the three isovelocities measured at the maximum and minimum knee joint angles. In addition the 14-point set included one extra (ω , θ , R_{exp}) point per isovelocity calculated at 50% of joint ROM while the 17-point set included two extra (ω , θ , R_{exp}) points per isovelocity compared to the 11-points, calculated at 33% and 66% of knee joint ROM, Table 1.

Regarding the lack of optimisation, we kindly refer you to our previous comment regarding our approach to selecting the number of points to fit to.

Line 273-278. I do not believe this is a rigorous way to carry out a cross-validation.

Authors' response:

We have reassessed this more comprehensively and added the following Lines 289-297

“Across the 14 subjects there was a spread of R^2 , RMSE and NRMSE scores and it was noted that in some case the shape of the curve could substantially deviate from the theoretical shape. To check for outliers a 14-choose-10 combinations was run on the RMSE values separately for 11-, 14- and 17-points. The combinations with the lowest RMSE score always had four of the same five subjects data missing. These combinations were 2.79 to 3.35 SD from the mean of all other combinations. The worst four had means across the 11-, 14-, and 17-point conditions that were between 3.69 and 6.41 SD from the group mean and were removed as outliers and the group data recalculated (Table 4).”

Figure 4. A figure including all data points and the curves should be included. It is very confusing to see only a single subject data point.

Authors' response: *We have combined Figures 4 and 5 into a single figure (Figure 5) that includes all data points and curves from the 11-, 14- and 17-point fits on two different subjects (2 and 4). Our aim is to demonstrate the juxtaposition between fitting the data on a subject whose R_{exp} values decrease with increasing ω and θ , and a subject with whose values R_{exp} values increase, as expected, with increasing ω and θ .*

Each subject's points were optimised to the surface fits on an individual basis so to show all the points would require as many separate curves per subject, per conditions to show them all.

Table 4. Why is a ~ 0.20 acceptable? What happens at higher speeds and velocities?

Authors' response: *We expanded more on this in discussion (lines 328-335)*

“When the number of points was reduced to 8, for both versions the initial R^2 and RMSE values were comparable to the 11-, 14-, and 17-point fits, but the NRMSE values were almost twice as large. There was a much greater range in surface

shapes with more deviating from the theoretical shape and some predicting negative ratio values whereas all measured values are positive. Paradoxically the R^2 values for these were good as even when the surface was unrealistic the $(\omega, \theta, R_{exp})$ points still fell on it as they were calculated near the extremes of ω and θ ranges with no midrange points to drag the middle of the surface above zero.”

Appendix B

Associate Editor Comments to Author (Dr Jonas Rubenson):

Comments to the Author:

Dear Dr. Voukelatos and co-authors,

As you will see, the review is generally positive. However, R1 raises some concerns that should be addressed. In particular, the reviewer's suggestions for Figure 1 and their comments about the concluding remarks can help improve your manuscript. I hope that you find the reviewer comments helpful I look forward to seeing a revised manuscript.

Authors' response: *We would like to thank both you and reviewer R1 for the constructive comments that have helped improve our manuscript. We have now amended Figure 1 and the concluding remarks to address these particular comments. We have also made further changes to the text to address all the remaining points raised in the review. We hope that we were able to do this successfully.*

Reviewer comments to Author:

Reviewer: 1

Comments to the Author(s)

The methods of the paper have been more clearly described. However the authors could put a bit more effort in making figure 1 more palatable (i.e. by adding respective graphs) and easier to understand.

The conclusion should clearly provide guidelines on the number of points (or experiments) that should be done to use the equation correctly. This does not come up clearly from the conclusion and should be included in the abstract too.

Also I feel that the title could be improve. The key message is to be able to use few experimental points as inputs to a novel equation to estimate H:Q ratio. The '3-D dimensional' description is not very helpful to me.

Authors' response:

Thank you for your useful comments and constructive critique of our work. We have updated figure 1 with some further descriptions within each step of the flow chart and by referencing the respective graphs in each of the steps. We have amended the abstract and conclusions to include a recommendation on the minimum number of points that should be done to use the equation correctly (Lines 14-15 & 458-463)

“Using 5 isometric points and 2 points at equally spaced angles from each of the 3 different isovelocities is recommended to give the best compromise between goodness of fit and minimal data points. This allows the smoothly curving theoretical shape to be produced when points across the measurements are well-behaved yet still allows for a divergence from this with an additional peak or trough or a negative surface gradient, for increasing ω and θ when some points are erroneous.”

and (Lines 479-480)

“... the $R_E(\omega, \theta)$ function can provide a fast and accurate description of the 3-dimensional H:Q_{fun} ratio profile of a test subject from 11 data points”

We have also amended the title of the manuscript to better reflect the key message of the work. The new title reads

“The Hamstrings:Quadriceps functional ratio expressed over the full angle-angular velocity range using a limited number of data points”

SPECIFIC COMMENTS

Line 98 - Please avoid the use of acronyms in this first sentence as it hinders the readability.

Authors' response: *We have amended the sentence as follows (Line 98):
"Raw torque-angular velocity-angle ($T-\omega-\theta$) datasets were used..."*

Line 102-103. What do the authors mean with 'test the robustness'? Is this a validation? If so what's the acceptable error and why?

Authors' response: *We used the term robustness to describe the effect of fitting the R_E function on different number of data points on the R^2 , RMSE and NRMSE values and that these values do not really change between fitting constraints. However, as this may indeed be taken to imply a validating procedure, we amended the text and substituted any references to "robustness" with the term "goodness of fit" (Lines 103-104, 243, 314-315)*

Line 194-195. How did the authors vary the combinations? What was the rationale for moving from linear to non-linear? I think this needs more explanation.

Authors' response:

The two functions were either linear combined by adding them, $R_E^C(\theta) + R_E^C(\omega)$, or non-linearly combined by multiplying the by a variable and then adding them, $\omega R_E^C(\theta) + \theta R_E^C(\omega)$, or by multiplying them, $R_E^C(\theta) \cdot R_E^C(\omega)$. We have re-written this particular paragraph to clarify the process of arriving at the R_E function (Lines 195-200).

"Equations (6) and (7) indicated that the two-variable $R_E(\omega, \theta)$ function would need to contain a combination of exponential and polynomial terms with respect to both ω and θ . Consequently, various linear, e.g. $R_E^C(\theta) + R_E^C(\omega)$, and non-linear, e.g. $\omega \cdot R_E^C(\theta) + \theta \cdot R_E^C(\omega)$ or $R_E^C(\theta) \cdot R_E^C(\omega)$ combinations of $R_E^C(\theta)$ and $R_E^C(\omega)$ were fitted on the $R_T(\omega, \theta)$ surface in order to determine $R_E(\omega, \theta)$. The best fit was obtained by the 6-parameter exponential function..."

Line 199 Perhaps I am missing something but how can the authors test the best fit using the same 11 subject? Should not this be tested on the 14 subject database? Perhaps the authors are referring to testing the robustness of the 17 points?

Authors' response: *The reduced parameter function was first fitted to the whole R_T surface for each one of the 11 subjects. Subsequently the fit was repeated but this time on a limited number of surface points (17), as the minimum needed to define the full R_T surface), not the whole surface. This paragraph has been amended to further clarify this (please see response to the comment below).*

Line 198-201. I am sorry but this paragraph is still too cryptic.

Authors' response: This paragraph has been amended to clarify that R_E was initially fitted to the whole R_T surface for each one of the 11 subjects and subsequently to only 17 surface points per subject (Lines 202-209)

“To test the ability of the $R_E(\omega, \theta)$ function to accurately reproduce the $R_T(\omega, \theta)$ the function was first fitted individually to each of the 11 $R_T(\omega, \theta)$ surfaces using the whole surface for the fit. Subsequently, to establish if it could accurately do so using a limited number of data points only 17 $(\omega, \theta, R_T(\omega, \theta))$ points from each of the 11 theoretical $R_T(\omega, \theta)$ ratio surfaces were chosen and $R_E(\omega, \theta)$ was fitted per subject to those points and compared to the whole surface fits. Seventeen $(\omega, \theta, R_T(\omega, \theta))$ points were used for the fit as the minimum number of points required to define each of $R_T(\omega, \theta)$ surfaces..”

Line 207 - I think should be equation 8.

Authors' response: Thank you for pointing this out. This has now been corrected (Line 215)

Line 462. How can the reader know what's the correct shape? I guess the reader would know what are the inputs (i.e. number of points) needed to use the proposed formula?

Authors' response: This has now been amended to clarify the expected shape of the graph (Lines 473-475)

“Although not an original aim of this study it was found that obtaining a shape for the whole surface that adheres to the fundamental muscle mechanical properties, rather than a value at any single point on the surface...”